

# PRIMAP-crf: UNFCCC CRF data in IPCC 2006 categories

M. Louise Jeffery[1], Johannes Gütschow[1], Robert Gieseke[1], and Ronja Gebel [1]

[1] Potsdam Institute for Climate Impact Research (PIK), Member of the Leibniz Association, P.O. Box 60 12 03, D-14412 Potsdam, Germany

*Correspondence to*: M. Louise Jeffery (louise.jeffery@pik-potsdam.de)

**Abstract.** All Annex I Parties to the United Nations Framework Convention on Climate Change (UNFCCC) are required to report domestic emissions on an annual basis in a 'Common Reporting Format' (CRF). In 2015, the CRF data reporting was updated to follow the more recent 2006 guidelines from the IPCC and the structure of the reporting tables was modified accordingly. However, the hierarchical categorisation of data in the IPCC 2006 guidelines is not readily extracted from the

reporting tables. In this paper, we present the PRIMAP-crf data as a re-constructed hierarchical dataset according to the IPCC 2006 guidelines. Furthermore, the data is organised in a series of tables containing all available countries and years for each GHG individual gas and category reported. It is therefore readily usable for climate policy assessment, such as the quantification of emissions reduction targets.

In addition to single gases, the Kyoto basket of greenhouse gases ($CO_2$, $N_2O$, $CH_4$, HFCs, PFCs, $SF_6$, and $NF_3$) is provided

according to multiple global warming potentials. The dataset was produced using the PRIMAP emissions module. Key processing steps include; extracting data from submitted CRF Excel spreadsheets, mapping CRF categories to IPCC 2006 categories, constructing missing categories from available data, and aggregating single gases to gas baskets. Finally, we describe key aspects of the data with relevance for climate policy; the contribution of $NF_3$ to national totals, changes in data reported over subsequent years, and issues or difficulties encountered when processing currently available data. The

processed data is available under an Open Data CC BY 4.0 license, and available at http://doi.org/10.5880/pik.2018.001.

## 1. Introduction

Under the United Nations Framework Convention on Climate Change (UNFCCC), Annex I countries are required to report detailed GHG emissions inventories to the UNFCCC on an annual basis. All data are reported in the 'Common Reporting

Format' (CRF) Excel tables and accompanying National Inventory Reports (NIRs), which are made available on the UNFCCC website (UNFCCC, 2017c). In addition to those Parties listed in Annex I of the Convention (UNFCCC, 1992), Kazakhstan is considered a member of Annex I for the purpose of the Kyoto Protocol. Malta and Cyprus also report CRF data as members of the European Union, but are not listed under Annex I of the Convention.

Prior to 2015, CRF data were reported following guidelines published by the IPCC in 1996 (IPCC, 1996), which detail the methodologies that should be used to calculate the inventories. Since 2015, reporting follows revised IPCC guidelines published in 2006 (IPCC, 2006). The methodologies to calculate emissions are primarily based on combining activity levels and emissions factors. For example, how many cows does a country have (activity), and what are the typical methane

emissions per cow (emissions factor)? For each source, multiple methods of calculation are possible with increasing complexity from tier 1 to tier 3. In general, tier 1 methods use a basic calculation and default emissions factors, tier 2 incorporates country specific emissions factors, and tier 3 may include advanced modelling approaches alongside country specific emissions factors. Tier 3 approaches are generally considered more accurate but may require more information and are more difficult to calculate.

In addition to revisions in methodology and emissions factors, the IPCC 2006 guidelines (IPCC, 2006) and supplementary methods arising from the Kyoto Protocol (IPCC, 2013) updated the range of activities and gases covered, and the hierarchical categorisation of the data. Major changes included (1) Combining categories 2 (Industrial Processes) and 3 (Solvent and Other Product Use) to the new category Industrial Processes and Product Use (IPPU) and (2) combining the Agriculture and Land-Use, Land-Use Change, and Forestry (LULUCF) categories into one Agriculture, Forestry, and Other

Land-Use (AFOLU) category.

Although the more up to date methodological guidelines are now followed, the CRF reporting tables still closely resemble the structure of the previous version. For example, Agriculture and LULUCF emissions are still reported separately and top-level tables of fugitive emissions from fuels still reflect IPCC 1996 categories. The legacy effect is primarily a result of the negotiating process.

A consistent and complete hierarchical dataset according to the 2006 guidelines and categories allows (1) checks on data consistency to be made, and (2) comparison with other datasets. For these reasons, we have extracted and processed all reported CRF data and re-organised it to the IPCC 2006 categorisation (Fig. 1). Furthermore, we make the data available in an easily used, machine readable, comma-separated values (CSV) file with aggregate gas baskets according to multiple global warming potentials (GWPs). Each table contains data for all countries and years available, allowing for time series

analysis and comparison between countries.

## 2.   Methods

### 2.1.   General approach

Annual CRF data for all countries is available from the UNFCCC website (UNFCCC, 2017c). The CRF data for each

country consists of one file for every inventory year, currently at least 26 per country (1990 to 2015), with some countries



reporting additional years to include their Kyoto Protocol base years (1986-1989). Each file contains detailed data for separate categories and summary tables as required by UNFCCC decisions (UNFCCC, 2017a).

To prepare the PRIMAP-crf dataset, the necessary data is read, processed, and prepared using the PRIMAP emissions module (Nabel et al., 2011). The PRIMAP emissions module is a MATLAB (2017) based tool with a custom database

structure ('PRIMAPDB') that can handle conversion of GHG emissions data with different units and GWPs. Each table in the database consists of a country-year matrix and is defined in terms of an entity (e.g. $CO_2$), category (e.g. IPC4 (AFOLU)), class (e.g. TOTAL), type (e.g. NET), scenario (e.g. HISTORY), and source (e.g. CRF2017).

Data in the individual spreadsheets is first mapped to the PRIMAPDB table structure according to a specifications file. As the data is not reported directly against the IPCC 2006 categorisation, we define a set of categories that are specific to the

CRF data and are later used to build the IPCC 2006 categories from sub-categories. Categories in the PRIMAP emissions module but outside the main IPCC 2006 categorisation are denoted as "M" categories in the emissions module and in this paper (e.g. M0EL is "national total excluding LULUCF"). Not all data available in the CRF tables is read in to the PRIMAP database. The level of detail is determined based on two criteria; (1) data that is useful for policy assessment, or (2) data that is required to build complete categories from the bottom up. Upon read-in to the PRIMAPDB, data from individual CRF

tables is aggregated across countries and years to give tables for individual gases, categories, class, and type.

After raw data is read-in, the full IPCC 2006 category hierarchy is constructed by addition of sub-categories. Any directly read data from higher sub-categories is not replaced at this stage, rather the addition of sub-categories is used to fill in any gaps where data is not directly reported.

Next, 'gas baskets' are generated by summing across gases to generate tables with all HFCs (hydrofluorocarbons), PFCs

(perfluorocarbons), F-gases (HFCs, PFCs, $SF_6$, and $NF_3$), or Kyoto-GHG ($CO_2$, $N_2O$, $CH_4$, HFCs, PFCs, $SF_6$, and $NF_3$). Each gas basket is created from individual gases according to four different sets of GWPs - Second Assessment Report (SAR), Fourth Assessment Report (AR4), Fifth Assessment Report (AR5), and AR5 with carbon-cycle feedbacks (AR5CCF) (IPCC, 1995, 2007, 2014). Some gas species are included in later IPCC reports but not in earlier ones. Those gases for which a GWP estimate was not available at the time of publication of a particular report are assigned a GWP according to the next report in

which a GWP estimate was given (see Supplementary Material for GWPs used in PRIMAP-crf).

Finally, the data is checked and verified for consistency and accuracy against the original data. First, we verify that the data is internally consistent in that each category in the hierarchy is equal to the sum of its sub-categories. This check is performed across all gases and countries individually, and both for categories that were generated in post-processing and those that were directly read-in. For example, we verify that category 3, created as the sum of its sub-categories, is also equal

to the sum of the directly read Agriculture (MAG) and LULUCF (MLULUCF) categories. If the discrepancy between different category levels exceeds 0.01% for any country or year, that data is flagged and checked. Where possible, discrepancies are fixed and, where not, we note them in the results section below. Second, a manual check is performed against the raw tables and national inventory reports, particularly for the top-level categories (e.g. IPC0). Third, we compare

top-level categories with data downloaded directly from the UNFCCC data portal (UNFCCC, 2017b). Fourth, data for main categories and gases is plotted against data from previous years to check for major changes. The source of any major differences between data in different years is verified to ensure that it is a real change in the raw data and not an error in the read-in process.

5       ## 2.2.   AFOLU

The most challenging part of the IPCC 2006 hierarchy to construct from CRF data is the AFOLU category (3). This is because AFOLU data is still reported in two entirely separate sections (CRF Table 3. Agriculture and Table 4. LULUCF), but the IPCC 2006 categorisation integrates emissions from some Agriculture and LULUCF sources into the same categories.

Categories 3A and 3B are exclusively Agriculture and LULUCF respectively, but category 3C combines data from both sectors (Fig. 2). To construct the category 3C sub-categories, multiple PRIMAP 'M' categories were created and data read-in at its most detailed level to these categories. A full description of the mapping from CRF data to PRIMAP-crf categories can be found in the Supplementary Material.

### 2.3.   Bunkers and Multi-lateral operations

Emissions from international aviation and marine bunkers, and multi-lateral operations are included in the IPCC 2006 categorisation and are reported to the UNFCCC. However, emissions from these sectors are not included in the national totals for reporting or accounting purposes under the UNFCCC and are therefore also excluded in the PRIMAP-crf hierarchy and national totals. Instead, data for these sectors is provided in separate 'M' categories (Table 1). Only France, Slovenia, Sweden, and Ukraine currently report emissions from multilateral operations.

## 3.   Results

### 3.1.   National totals

The PRIMAP-crf dataset is intended for multiple uses. Two key examples include (1) comparison of national total emissions between Parties (Fig. 3) and (2) analysis of the share of emissions from each major category within the national total (Fig. 4). In Fig. 4 we also show the additional emissions from international bunkers and how these compare to national totals.

### 3.2.   Changes in reported data through time

Annual updates to the data reported by countries in their national inventories do not only extend the time series but also include updates to previous data years according to revisions in source data, updates in methodologies, and updates in

reporting guidelines. The extent to which the reported data changes varies by sector; energy emissions have a low uncertainty compared to LULUCF emissions and accordingly do not undergo such significant annual modifications.

We do not identify the reason for all updates to historic time series but isolate significant changes related to the transition to new reporting guidelines. We find that the 1990 to 2015 reported national total emissions (excluding LULUCF) for Monaco,

Croatia, Hungary, and Cyprus all dropped significantly (>5% on average over 1990-2012) between the 2014 and 2017 releases, and that Russia and Malta's reported emissions increased by >5% on average over 1990-2012. Average reported total emissions (excluding LULUCF) for Australia, Bulgaria, France, Lithuania, Portugal, and Kazakhstan all changed by 3-5% over the same period between the 2014 and 2017 versions.

In the LULUCF sector, we identify three countries for which the new guidelines introduced substantially different overall

emissions levels - Iceland, Ireland, and the Netherlands (Fig. 5). Updates to emissions factors for drained organic soils and methods for accounting for emissions associated with peat extraction result in much higher reported LULUCF emissions from Iceland and Ireland. LULUCF emissions from the Netherlands are also significantly higher, but the increases are distributed across several land-use categories and not so readily attributable to a single source. The new guidelines also introduced significant changes to the LULUCF time series from many other countries.

## 3.3.  Data Issues

The current format of the CRF data allows for some flexibility in reporting for countries. As a result, it was not possible to obtain data for all categories for all countries. In addition, the detailed checks revealed some issues in the raw data where data reported in one table was not consistent with data reported in another table. Here we briefly describe those issues and how it impacts the final dataset. We first describe structural issues with the data tables and then some country specific

inconsistencies in the reported data.

Country data reported to the UNFCCC undergoes mandatory review by an Expert Review Team according to the Guidelines for review under Article 8 of the Kyoto Protocol (most recently updated in Decision 13/CP.20). Expert review reports are made available on the UNFCCC website (UNFCCC, 2016). Many of the country specific issues identified here may be picked up in the UNFCCC Expert Review process and corrected in future updates of the CRF data. If so, future updates of

PRIMAP-crf will also incorporate these corrections.

### 3.3.1.   Table construction

**Fugitive emissions from solid fuels**

Some sub-categories of emissions from category 1B1A 'Coal mining and handling' are not explicitly and separately reported in the current CRF tables and therefore could not be included as separate categories in PRIMAP-crf. These sub-categories

are 1B1B 'Uncontrolled Combustion, and Burning Coal Dumps' and 1B1A14 'Flaring of drained $CH_4$ or Conversion of $CH_4$ to $CO_2$'.



Category 1B1A is reported in table 1.B.1 of the CRF tables, which includes lines for '1.B.1.b. Solid fuel transformation' and '1.B.1.c. Other (please specify)'. Additional information indicates that fugitive emissions from coke and charcoal production may be included in the 1.B.1.b line, and that the 'Other' category may be used for 'reporting any other solid fuel related activities resulting in fugitive emissions, such as emissions from waste piles.' If emissions from either IPCC 2006 category 1B1B or 1B1A14 are calculated and reported in either of these lines, then they are included in the national totals (category 0) but cannot be reported explicitly as separate categories in the PRIMAP-crf data.

A further issue with the CRF tables for category 1B1 is that two countries (Australia and United Kingdom) report $N_2O$ emissions from category 1B1 in the summary tables (e.g. Table 1s2 and Table 10s4) but it's not possible to include the details on $N_2O$ emissions from this category in the Table 1.B.1, which only has columns for $CO_2$ and $CH_4$. Because the CRF data for category 1B1 is read from Table 1.B.1 and not the summary tables, a discrepancy occurs when comparing the category 1B total for $N_2O$ with the sub-categories 1B1, 1B2, and 1B3 for Australia and the United Kingdom. The sub-totals are up to 0.2% too low for the United Kingdom, and 0.9% too low for Australia due to the missing category 1B1 data.

**IPC1A2 sub-categories for $N_2O$, $CH_4$, and $CO_2$**

It is not mandatory to report all sub-categories of category 1A2 (fuel combustion for manufacturing industries and construction) explicitly in CRF tables. Sub-categories of 1A2 are reported in CRF table 1.A(a)s2 with explicit lines for 1A2A through 1A2F (Fig. 6) and optional reporting under 'Other' for categories 1A2G through 1A2M. Bulgaria, Germany, Estonia, France, Ireland, Italy, and Japan do not fully report these additional sub-categories and the sum of their 1A2 sub-categories in PRIMAP-crf is therefore not equal to the category 1A2 total. However, for category 1A2 and higher levels in the hierarchy, the data is complete and correct.

### 3.3.2. Country specific

Some data issues are not structural issues with the CRF table set-up but result from the way in which individual countries report the data. Here we describe inconsistencies found for individual countries during the read-in and testing process.

**Germany**

Germany does not report $CH_4$ emissions from drainage and rewetting of soils on settlements land in table 4(II) or in table 4. The data is listed as 'IE' (Included Elsewhere) but it is not apparent where. The documentation notes suggest that it is included in table or category 4E. However, CRF Table 4.E is for $CO_2$ only, so it cannot be reported there. As a result, the PRIMAP-crf tests that compare category 3 and category 3 sub-categories with the directly read-in LULUCF data fail for Germany.

Germany also failed tests that check whether the national totals of HFCs, PFCs, and F-gases are equal to the sum of the main sub-categories, in this case just category 2. The tests only fail for the final year (2015) and for AR4 GWPs. This indicates a discrepancy in the original data. Inspection of Table 10s5 shows that the value reported for 2015 for the HFC and PFC



baskets is approximately 1000 times higher than that reported in 2014, suggesting a units error in reporting. We expect this error to be corrected in future submissions. The spurious data points are corrected in post-processing and not present in the PRIMAP-crf source.

**Kazakhstan**

In the raw Excel file (dated 17-11-2017) for 1993, the $CO_2$ emissions factor for gas/diesel oil for domestic navigation is too high by a factor of $10^6$, resulting in emissions for the category 1A3D2 that are too high by the same magnitude for that year. In the PRIMAP-crf dataset we manually replace this data point and use the corrected data to recalculate the higher-level categories. The emissions factor was corrected from $74.1 \times 10^6$ to 74.1 (t/TJ), resulting in 40.502 kt $CO_2$ for category 1A3D2 from Kazakhstan in 1993, which fits with the full time-series. This correction is applied to all affected higher level categories

and gas baskets in the final PRIMAP-crf dataset.

**Lithuania**

$N_2O$ emissions in category 3 (AFOLU) for Lithuania do not equal the sum of reported emissions under Agriculture and LULUCF. This discrepancy occurs because total category 3 emissions are calculated from the sum of sub-categories (Fig. 2) and $N_2O$ data is not completely reported in LULUCF sub-categories for Lithuania. The issue occurs with sub-category 3C4

(Other land), which is partially reported in CRF table 4(III) and read-in under PRIMAP category M3C4LUBOTH. However, the total N2O LULUCF emissions from Other Land reported in summary table 4 are higher than those reported in table 4(III). Table 4 is also not internally consistent and it is not clear from the CRF tables what the source of the additional emissions is.

As a result, the PRIMAP-crf data fails tests for $N_2O$ that compare category 3 with LULUCF. The LULUCF and national

total (category 0) values are correct. The error propagates to the Kyoto-GHG basket, which includes $N_2O$.

**Luxembourg**

$N_2O$ emissions data for category 3 (AFOLU) are also incorrect for Luxembourg for similar reasons to Lithuania. Data is reported under 'Other land' in CRF Table 4 that is not reported in any other table, including Table 4(III) where it would be expected. As the data is not read-in to a sub-category, the resulting aggregated category 3 is not consistent with the directly

read-in LULUCF category. As with Lithuania, category 3 is incorrect, but the LULUCF and national total categories are correct. The error propagates to the Kyoto-GHG basket for category 3, which includes $N_2O$.

**Norway**

Norway fails a test for $N_2O$ emissions that compares the directly read-in LULUCF total to the sum of its components. The error averaged over all years is 0.017%. The source of the discrepancy is unclear but CRF Table 4 is internally inconsistent

suggesting some missing data in sub-categories. Because the error is small, tests on higher level categories and aggregate gas baskets are all passed.



**Sweden**

For the year 2015, Sweden reports fugitive emissions from Oil production in Table 1.B.2, item 1.B.2.a.Oil. However, for $CH_4$ and $CO_2$, no data is reported in the subsectors. For $N_2O$, no data is reported at all in Table 1.B.2, but data is reported in

Table 1s2, item "2. Oil and natural gas and other emissions from energy production".

A further problem in Sweden's CRF reporting is an inconsistency between tables. In Table 1.s.2, item "c. Other (as specified in table 1.B.1)" $N_2O$ emissions are reported. However, Table 1.B.1 does not allow reporting of $N_2O$ emissions and thus the sum obtained from Tables 1.B.1 and 1.B.2 does not coincide with the data read from Table 1.s.2.

In category 2 (Industrial processes and product use), 0.36 kt of methane emissions are reported for 2015 while the data

available for subsectors only sums to 0.01 kt. There are discrepancies in subcategories: the category 2B10 (chemical industry - other) has no data reported for $CH_4$ in Tables 2(I)s1 and 2(I).A-Hs1 but a subcategory (other inorganic chemical products) contains data. But probably most important is that nothing is reported in Table 2(I).A-Hs1, item "2.H.1 Pulp and paper" where the majority of $CH_4$ emissions for category 2 was reported for 2014.

Consequently, CRF data for Sweden fails tests that compare category 1.B to its subcategories for $CO_2$, $CH_4$, $N_2O$ and the

Kyoto-GHG basket, as well as tests that compare category 2 to its subcategories for $CH_4$. The errors propagate to the Kyoto-GHG basket.

### 3.4. $NF_3$

For the second period of the Kyoto Protocol (2013-2020), $NF_3$ emissions must be reported and accounted in all historical

data. Emissions of $NF_3$ are primarily from the chemical and electronics industries (CRF Tables 2(I)s1 and 2(I)s2). Despite its relatively high 100-year GWP of 16,100 (Myhre et al., 2013), the share of emissions from $NF_3$ is low; the average annual contribution of $NF_3$ to total Kyoto-GHG (AR4 GWPs) since 1990 ranges between 0.000032% for Canada and 0.053% for Japan (only 13 countries report $NF_3$ emissions). Japan and the USA are the two major emitters of this gas , but $NF_3$ emissions from Japan reduced by more than half between 2013 and 2015.

### 3.5. Indirect $N_2O$ and $CO_2$ emissions

Another addition for the second commitment period of the Kyoto Protocol is that countries may now also report indirect $N_2O$ and $CO_2$ emissions from additional sectors, whereas previously only indirect $N_2O$ from agriculture was explicitly listed in the reporting tables. Indirect emissions are greenhouse gases that result from secondary chemical reactions of precursor substances. Indirect $CO_2$ emissions are the result of oxidation of carbon containing non-$CO_2$ gases, such as $CH_4$, CO, and

non-methane volatile organic compounds (NMVOC). Indirect $N_2O$ emissions have two primary sources; (1) volatilisation



and subsequent atmospheric deposition of $NH_3$ and NOx, and (2) leaching and runoff of added N to water systems, such as from the use of nitrogen fertilisers in agriculture (IPCC, 2006).

The UNFCCC COP Decision 24/CP.19 states that: "29. In addition, Annex I Parties should provide information on the following precursor gases: carbon monoxide (CO), nitrogen oxides ($NO_X$) and non-methane volatile organic compounds (NMVOCs), as well as sulphur oxides ($SO_X$). Annex I Parties may report indirect $CO_2$ from the atmospheric oxidation of $CH_4$, CO and NMVOCs. Annex I Parties may report as a memo item indirect $N_2O$ emissions from other than the Agriculture and LULUCF sources. These estimates of indirect $N_2O$ should not be included in national totals. For Parties that decide to report indirect $CO_2$ the national totals shall be presented with and without indirect $CO_2$." (UNFCCC, 2017a)

Indirect $N_2O$ emissions from the AFOLU sector are therefore included in the national totals (category 0), but indirect emissions from other sectors are not. In PRIMAP-crf, category 0 'national total' also excludes indirect $CO_2$ emissions. In PRIMAP-crf, national totals including indirect $CO_2$ are included as category 'M0INDRCT', and separate indirect $N_2O$ and $CO_2$ emissions are contained as categories M0N2OIND and M0CO2IND respectively.

As reporting of indirect emissions is optional, data is not available for all countries. Countries currently reporting indirect $N_2O$ emissions from non-AFOLU sectors are Belarus, Bulgaria, Czech Republic, Denmark, Finland, Italy, Norway, Romania, Sweden, Switzerland, and the United Kingdom. Countries currently reporting indirect $CO_2$ emissions are Canada, Czech Republic, Denmark, Finland, Japan, Latvia, Netherlands, Portugal, and Switzerland.

Note that it is necessary to distinguish between different usages of the term 'indirect emissions'. The indirect emissions referred to here are emissions resulting from subsequent reactions of GHG precursors. However, businesses, industry and the IPCC also use indirect emissions to refer to emissions that result from the activities of that entity, but that are not under their direct control (GHG Protocol, 2017). For example, a business may include the emissions associated with heating its offices as indirect emissions.

## 4. IPCC 1996 categories

The 1996 categories are still useful for comparison and combination with older data sources. We therefore created a mapping of higher level 2006 categories to 1996 categories to enable the use of the PRIMAP-crf dataset in such cases, especially for the forthcoming v1.2 of the PRIMAP-hist historical time-series dataset (Gütschow et al., 2016). As the new CRF tables are very similar to the old IPCC 1996 CRF tables, building the 1996 categories can be achieved with reading a few additional items from the tables. Table 2 shows which IPCC2006 categories are used to create each IPCC1996 category. Some additional items from the CRF tables are needed and read into "M" categories. The categorical detail is selected according to those required for the PRIMAP-hist dataset (Gütschow et al., 2016). Consequently, we create the main categories (1–7) for all gases and additionally the subcategories of categories 1 and 2 for $CO_2$. National totals under the two sets of guidelines

differ for two primary reasons: (1) the IPCC1996 guidelines do not include all sources covered by the 2006 guidelines. At the level of detail we employ for the conversion, this only affects the carbon capture and storage (CCS) related emissions reported in the new category 1C, which are not included in 1996 categories. (2) For the creation of IPCC 1996 categories 1 and 2 we need detailed subcategories which are not reported in full detail by all countries. In other cases, emissions are not

reported consistently across the different tables. This means that sometimes we cannot perfectly create the IPCC 1996 categories. Both effects are very small.

## 5.   Data Availability

The dataset is available from the GFZ Data Services under doi:10.5880/PIK.2018.001 (Jeffery et al., 2018). When using this dataset, or one of its updates, please cite this paper and the precise version of the dataset used. Please also consider citing the

original UNFCCC source (UNFCCC, 2017c) when using this dataset.

## 6.   Conclusions

We provide a processed dataset of the CRF data reported to the UNFCCC in an easily used format (PRIMAP-crf) and accompanied by a detailed description of its derivation. Two key facets of the processed data are that (1) data for all countries and years is readily accessed for a given gas and category, and (2) the data is categorised in a consistent hierarchy

according to the IPCC 2006 guidelines. In addition, we also provide data in top-level categories under the IPCC1996 categorisation.

PRIMAP-crf data (Jeffery et al., 2018) is provided under an open CC BY 4.0 licence and can be accessed at http://doi.org/10.5880/pik.2018.00. The data is provided in a machine-readable format (CSV) for bulk download. As such it can be used easily and flexibly, and is not tied to any specific software or operating system.

Some structural issues with the current table format have been identified and country specific issues with reported data have been highlighted.

## 7.   Author Contributions

L.J. wrote the manuscript with contributions from J.G. and R. Gieseke. L.J., J.G., and R. Gebel processed and prepared the

dataset. L.J. and J.G. maintain the PRIMAP emissions module.





## 8. Acknowledgements

The authors acknowledge and appreciate funding by the German Federal Ministry for the Environment, Nature Conservation and Nuclear Safety (11_II_093_Global_A_SIDS_and_LDCs, 16_II_148_Global_A_Impact). We wish to thank previous members of PRIMAP emissions module group - Claudine Chen, Malte Meinshausen, and Julia Nabel - for their development
of the PRIMAP emissions module and processing earlier versions of the CRF data.

The development of this dataset and the paper made use of several open source software tools; we acknowledge and thank the developers of colorbrewer2.org and MacDown. The tree diagram figures were constructed based on D3 code by Mike Bostock.

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



## 10. Figures

**Figure 1: Top three levels of the IPCC 2006 categorisation. Filled circles indicate that further sub-categories exist. Orange circles indicate that the category is generated by aggregation of sub-categories in PRIMAP-crf. A fully expandable and explorable version of this graphic is available at https://www.pik-potsdam.de/paris-reality-check/primap-crf/.**





**Figure 2: AFOLU categorisation. Agriculture only (red) and LULUCF only (green) categories are aggregated to give combined categories (purple). Some of the data that is provided as Agriculture or LULUCF only does not map directly to an IPCC 2006 category. Additional PRIMAP 'M' categories are therefore created for this data and integrated into the hierarchy. Filled circles indicate categories for which further sub-categories are available in the full dataset. An interactive version of this figure is available at https://www.pik-potsdam.de/paris-reality-check/primap-crf/.**





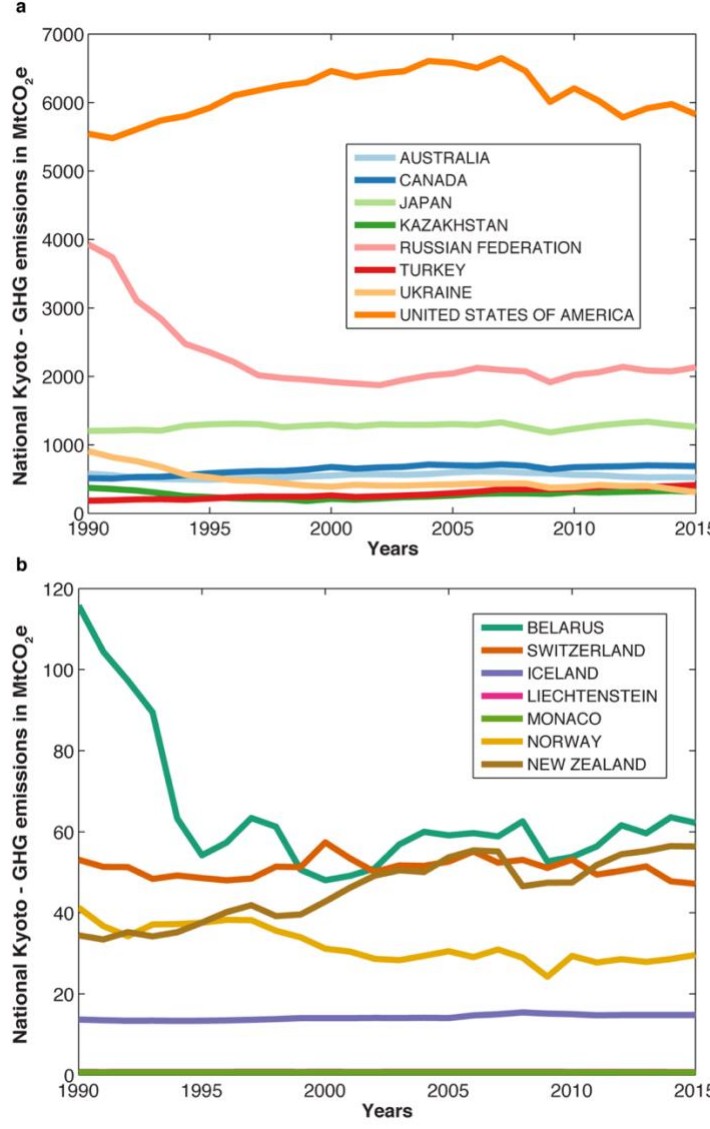

**Figure 3: Time series of national total emissions (including LULUCF) as reported in CRF tables for larger emitters (a) and smaller emitters (b).**

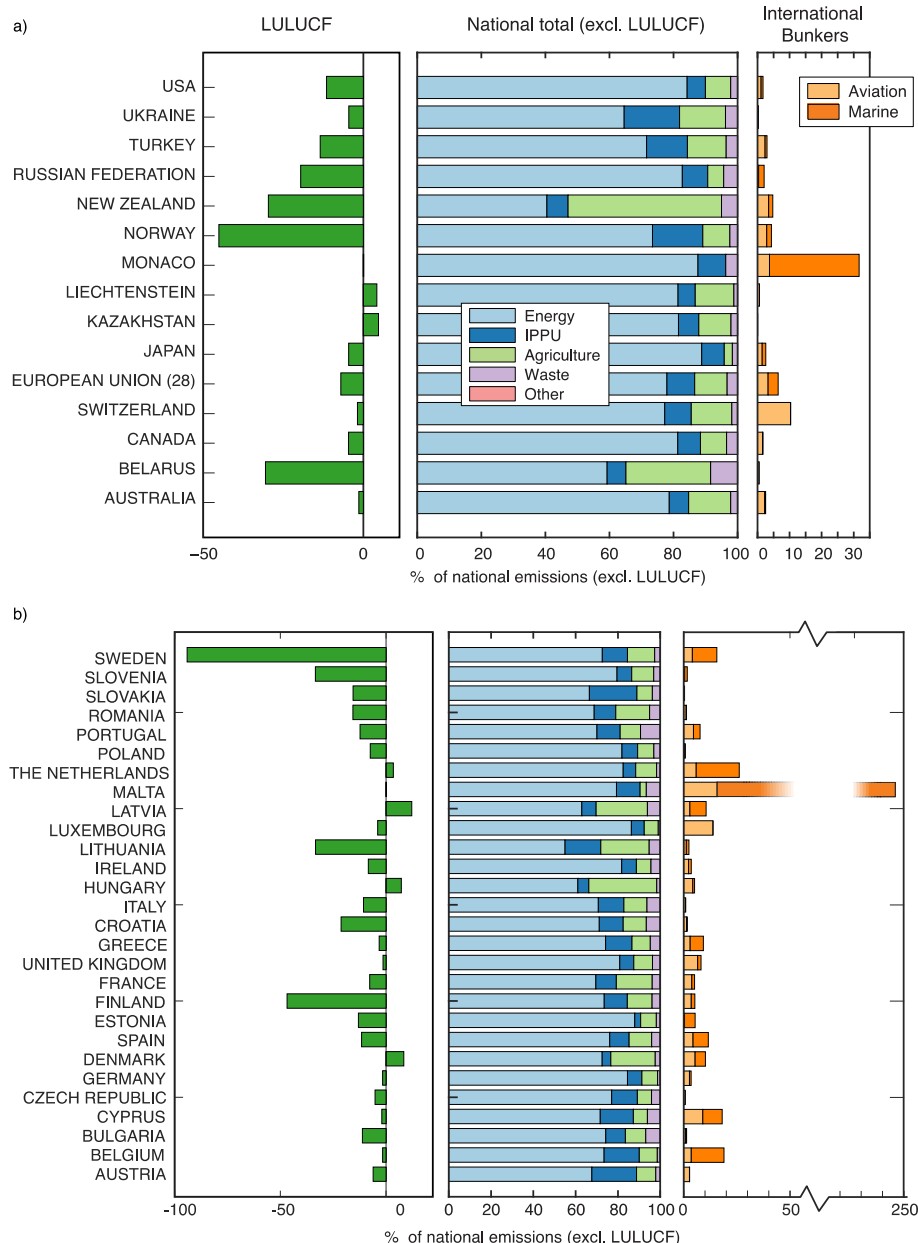

**Figure 4: Shares of sectors in the national total emissions, excluding LULUCF and international bunkers. For comparison, reported emissions from LULUCF are shown as % of national total (green, LHS) and reported emissions for international bunkers (orange, RHS). (a) Annex I countries and Kazakhstan, with the European Union presented as a group. (b) Same as (a) but for individual EU countries. Note the break in scale for international bunker emissions from Malta.**



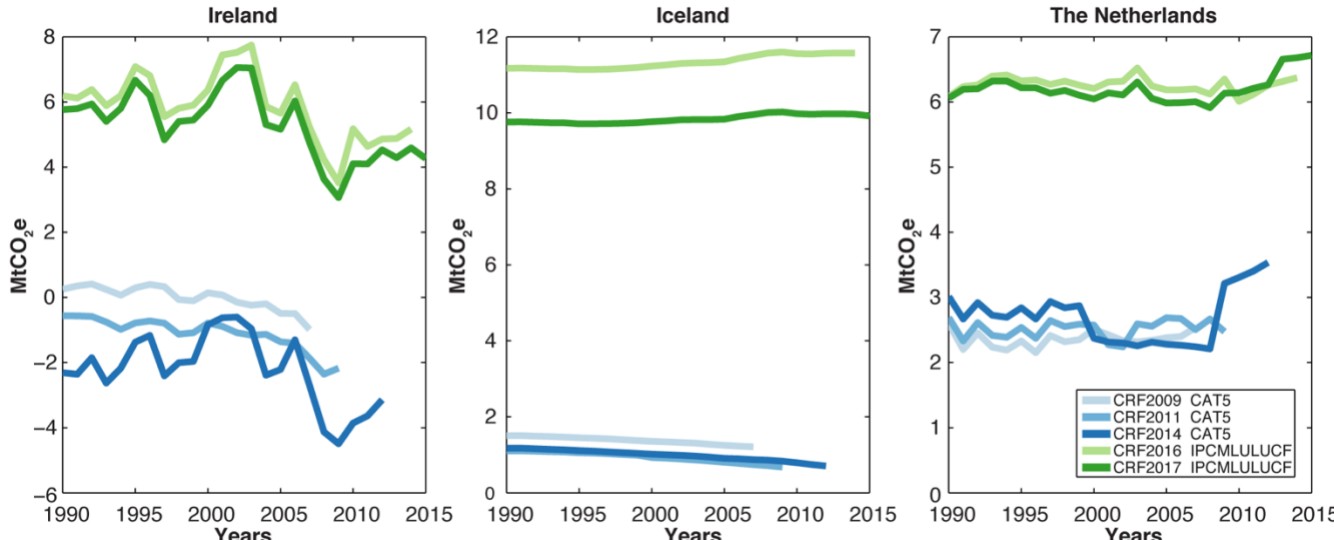

**Figure 5: Impact of new reporting guidelines (used from 2015 onwards) on reported Kyoto-GHG emissions in the LULUCF sector from Ireland, Iceland, and the Netherlands. Due to historic reporting conventions, Kyoto-GHG emissions in this figure are calculated using GWPs from the IPCC SAR (IPCC, 1995).**





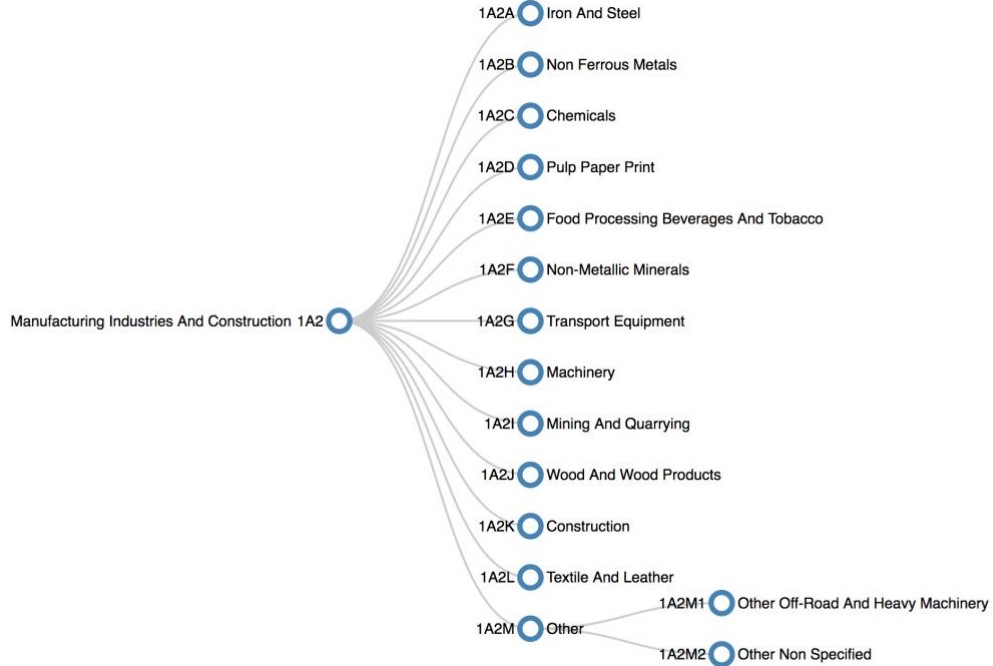

**Figure 6: Full category 1A2 sub-categories. For some countries, only sub-categories 1A2A to 1A2F are reported as sub-categories and categories 1A2G to 1A2M are reported together as 'other'. PRIMAP-crf contains the detailed categories where available.**



## 11. Tables

| Emissions source | 'M' category code |
|:---:|:---:|
| International Bunkers | MBK |
| International Aviation | MBKA |
| International Marine | MBKM |
| Multilateral Operations | MMULTIOP |

**Table 1: PRIMAP-crf category codes for emissions from international bunkers and multi-lateral operations. These codes are used in the data repository.**





| Gas | IPCC 1996 Category | Built from |
|---|---|---|
| CO$_2$ | CAT1 | CAT1A + CAT1B |
| | CAT1A | IPC1A |
| | CAT1B | CAT1B1 + CAT1B2 |
| | CAT1B1 | IPC1B1 + IPC1B3 |
| | CAT1B2 | IPC1B2 |
| | CAT2 | CAT2A + CAT2B + CAT2C + CAT2D + CAT2G |
| | CAT2A | IPC2A + IPC2B7 + IPC2D4 |
| | CAT2B | IPC2B − IPC2B7 |
| | CAT2C | IPC2C |
| | CAT2D | IPC2H1 + IPC2H2 |
| | CAT2G | IPC2H3 + IPC2H4 |
| | CAT3 | IPC2D1 + IPC2D2 + IPC2D3 |
| | CAT4 | IPCMAG |
| | CAT5 | IPCMLULUCF |
| | CAT6 | IPC4 |
| | CAT7 | IPC5 |
| CH$_4$ and N$_2$O | CAT1 | IPC1 − IPC1C |
| | CAT2 | IPC2 − CAT3 |
| | CAT3 | IPC2D1 + IPC2D2 + IPC2D3 + IPC2G3 |
| | CAT4 | IPCMAG |
| | CAT5 | IPCMLULUCF |
| | CAT6 | IPC4 |
| | CAT7 | IPC5 |
| HFCs, PFCs, SF$_6$ | CAT2 | IPC2 |

**Table 2: Creation of IPCC1996 categories from IPCC2006 categories and additional items from the CRF tables. 'CAT' is used to indicate IPCC1996 categories, and 'IPC' is used for IPCC2006 categories, both here and in the PRIMAP emissions module.**



## 12. Appendix

The following items can be found in the supplementary material to this paper.

5    1. **PRIMAP-crf-IPCC2006-category-codes.csv** A full documentation of the IPCC category codes used in PRIMAP and their meaning. Also include the PRIMAP 'M' categories used for reading CRF data and building categories.

2. **PRIMAP-crf_2018v1.0_categories.json** This JSON file shows which CRF table was used to build each category in the IPCC 2006 hierarchy. This file also provides the basis for figure 1.

3. **CRF_tables_key_IPCC2006.xls and CRF_tables_key_IPCC2006.csv** Similar to above, this file outlines the mapping from CRF tables to IPCC 2006 categories for the PRIMAP-crf source. The two files are identical except that the Excel version is formatted for easy reading, the CSV is unformatted.

15    4. **PRIMAP-crf_figure_codes_and_data** This folder contains the data (JSON) and code (HTML) underlying figures 1 and 2 of the main manuscript. Opening the HTML code in your browser (Firefox recommended) should enable you to explore the full hierarchy. An internet connection is required to retrieve the D3 library.

5. **PRIMAP_EMMOD_GWPs.xls and PRIMAP_EMMOD_GWPs.csv** These two files contain the 100-year global
20   warming potentials of individual gases used to generate the greenhouse gas baskets in the PRIMAP-crf dataset. The Excel file contains additional information about which IPCC Report each GWP value was derived in for those gases that were not included in the earlier reports.