# Peer review of "PRIMAP-crf: UNFCCC CRF data in IPCC 2006 categories"

_Earth System Science Data, 2018_

## Referee Comment (RC1) · Anonymous Referee #1 · 19 Mar 2018

Review ESSD-2018-3

Excellent overall!  Good description, good links to supplements and graphics, much needed step in the overall emissions quantification and validation process.  Easy to access and use the source date files.  Strong recommendation for publication in ESSD.

A few comments, mostly by way of suggestion:

Page 2 line 29 - Until this point the manuscript has carefully addressed only Annex I countries with a few explicit additions. Here, apparently, we switch to CRF data for "all" countries? Because most Annex II countries do not report CRF, this sentence must still refer only to Annex I? Worthy of clarification or, earlier in the text, emphasis that all subsequent discussion refers only to Annex I?

Page 3 line 6 - Here the reader confronts the short-hand acronym IPC4 which we understand if we have read the legend for Table 2 but otherwise we have not had explanation in the text?

Page 5 line 21 - Not until later in this paragraph do the authors clarify that the expert review occurs at the UNFCCC level.  At the start of the paragraph a reader does not know whether expert review happens at the country level or the international level.

.csv tables work well, with clear organisation and good explanations.  For some categories, e.g. the rarer HFCs, the table includes a strange mixture of formats, e.g. 0.000503 in one row but 1.2E-06 two rows below.  Do these differences arise from the CFRs (because each type of format tends to occur consistently across a row) or from PRIMAP?  Do we need a more consistent data formatting for these very small numbers?  In terms of GWP, not worth the effort?

Kyoto bucket data (e.g. KYOTOGHGAR4) occur mixed in with the individual gas data?  Although presentation of all data in one table with uniform formatting may prove necessary, the present organisation presents a substantial challenge to users (countries?) who might like to check specific PRIMAP outcomes e.g. where bucket total differs from sum of individual concentrations. Even if duplicative, should we have the bucket to individual components comparison as a separate table?  Or the authors could build in a summary line before or after each bucket group, showing cumulative individual concentrations for comparison?  We read about these discrepancies in the text e.g. in Section 3 but nowhere do we see a figure, which suggests to this reader that these discrepancies remain obscure except to the authors?  Not worth the effort if the discrepancies remain minor but some countries will want to see?

In Figure 4, left panel LULUCF appears on rough estimate about cumulative sink of  20% or so. From global carbon budget also in ESSD, for decade 2007 to 2016, fossil fuel emission roughly 10 GtC with a net land impact (sink minus sources) of very roughly 2 GtC, so again roughly 20%. Thinking about validation, for which these authors address only the PRIMAP product vs the original CFRs.  But they could perhaps validate their PRIMAP outcomes against carbon budget because the latter uses several sources in addition to the UNFCCC reports used here?  Even a brief mention of validation in text would strengthen reader confidence in their PRIMAP outcomes?

Also in Figure 4, right hand panel about bunkers, we should interpret this more as who reports and who doesn't rather than as actual emission numbers?

---

## Referee Comment (RC2) · Anonymous Referee #2 · 12 May 2018

Thanks for the excellent paper and data. I have only minor comments, as the paper presents a well done analysis, including quite some technical details. The transparency of the results is very good.

Minor comments: 1/ I would also present the impact of the new reporting guidelines on the emissions relative to 1990 levels (of BY emissions), including and excluding LULUCF, in a table. The %-changes are presented on page 5, the second paragraph, but I would also present an overview in a Table.

2/ Now the results of three countries are highlighted. I would also recommend to include a few additional other countries, as the current figure brings these countries on the spotlight.

---

## Referee Comment (RC3) · Anonymous Referee #3 · 17 May 2018

I think that this manuscript is well written and very useful for analyzing the large emission data and check possible errors in the reported data. I recommend this manuscript for publication.

---

## Referee Comment (RC4) · Anonymous Referee #4 · 22 May 2018

**Referee review for essd-2018-3**

Overall, this is a great study, very detailed and sophisticated. Thanks for such a good work. This data can be used for many advanced research and publications. Recommended to be published in ESDD after few, but not major clarifications.

In the line 27$^{th}$, it is stated that "Kazakhstan is included in the data set as a member of Annex I countries for the purpose of the Kyoto Protocol". What does this mean? Kazakhstan is considered as Annex-I because it ratified the Kyoto Protocol? If this is the reason, many other non-Annex I countries ratified Kyoto Protocol as well. Why only including Kazakhstan? Could you please clarify this little bit more?

Also, in the following sentence it is stated that Malta and Cyprus are reporting their emissions as members of European Union. Can you provide some reference for this, where do you get such information?

Lastly, at the data description file on page 2, it is stated there are 3 .csv files in the date sets. However, there are only 2 csv files in addition to two other text files. The following file is not there: "PRIMAP-crf-IPCC2006-category-codes.csv" Could you please include this file as well.

Please also check the correction of the measurements, there are several variables/units, a mistake easily can be made. Since I am not able to go over each calculation in details, I can only suggest you making sure that there is no mistake in the calculations.

After these corrections and clarifications, I recommend the study to be published in ESSD.

---

## Author Comment (AC1) · 29 May 2018

The supporting file "PRIMAP-crf-IPCC2006-category-codes.csv" is currently missing from the data repository. We apologise for missing this in the original file upload.

When the manuscript is revised, this category code file will be added to the repository, along with a revised dataset.

Until then, the category code file can be accessed in the Supplement to the manuscript that can be downloaded from the paper's main ESSDD page.

---

## Author Comment (AC2) · 5 Jul 2018

**Response to reviewers**

**PRIMAP-crf: UNFCCC CRF data in IPCC 2006 categories**

M. Louise Jeffery1, Johannes Gütschow1, Robert Gieseke1, and Ronja Gebel1

1 Potsdam Institute for Climate Impact Research (PIK), Member of the Leibniz Association, P.O. Box 60 12 03, D 14412 Detadem Commonly

D-14412 Potsdam, Germany

Correspondence to: M. Louise Jeffery (louise.jeffery@pik-potsdam.de)

**Overview**

We thank the editor and four anonymous reviewers for their time and constructive comments on the manuscript and data. In response to these comments we have made some edits to the manuscript and added some additional supporting material. A tracked changes version of the manuscript can be found at the end of this document.

In addition to the changes made in response to reviewer comments we have

- (1) updated figure 3 because the EU was missing in the original version of the paper. Some re-formatting was required to include the legend but the figure is otherwise unchanged.
- (2) Updated the data description in response to feedback from collaborators and in anticipation of releasing updated versions of the dataset (Page 2, Line 29 and Section 5)

Below are specific responses to the reviewers comments, with reviewer comments in black and author responses in blue.

Received and published: 19 March 2018

Excellent overall! Good description, good links to supplements and graphics, much needed step in the overall emissions quantification and validation process. Easy to access and use the source date files. Strong recommendation for publication in ESSD.

**Thank you for your positive feedback.**

A few comments, mostly by way of suggestion:

Page 2 line 29 - Until this point the manuscript has carefully addressed only Annex I countries with a few explicit additions. Here, apparently, we switch to CRF data for "all" countries? Because most Annex II countries do not report CRF, this sentence must still refer only to Annex I? Worthy of clarification or, earlier in the text, emphasis that all subsequent discussion refers only to Annex I?

You are correct, here 'all' refers to all Annex I countries and Kazakhstan. We have reworded the text here and elsewhere to say 'all reporting countries'. (Page 3 | Line 3 ; Page 5 | Line 25 ; Page 9 | Line 26 ; Page 11 | Line 8)

We assume that here you mean "non-Annex I" countries, that do not have the same reporting requirements under the Kyoto Protocol or Convention as the Annex I countries. All Annex II countries are also members of Annex I, but additionally have support obligations under Articles 2.3-2.5 of the Convention.

Page 3 line 6 - Here the reader confronts the short-hand acronym IPC4 which we understand if we have read the legend for Table 2 but otherwise we have not had explanation in the text?

Thank you for pointing this out, we have added (Page 3 | Line 10) a brief explanation that 'IPC' indicates the categorisation system used. 'IPC' and the difference to 'CAT' are then further explained later in the text (Page 10| Line 19-21) and in the caption for Table 2.

Page 5 line 21 - Not until later in this paragraph do the authors clarify that the expert review occurs at the UNFCCC level. At the start of the paragraph a reader does not know whether expert review happens at the country level or the international level.

We have reworded the first sentence of the paragraph to indicate that the Expert Review occurs under the UNFCCC. It now reads:

"Reported country data undergoes mandatory review under the UNFCCC by an Expert Review Team according to the Guidelines for review under Article 8 of the Kyoto Protocol (most recently updated in Decision 13/CP.20)." (Page 5 | Line 29)

For those interested, further information on the Expert Review Process can be found on the UNFCCC website - https://unfccc.int/index.php/process/transparency-and-reporting/reporting-and-review-under-the-convention/greenhouse-gas-inventories-annex-i-parties/review-process

.csv tables work well, with clear organisation and good explanations. For some categories, e.g. the rarer HFCs, the table includes a strange mixture of formats, e.g. 0.000503 in one row but 1.2E-06 two rows below. Do these differences arise from the CFRs (because each type of format tends to occur consistently across a row) or from PRIMAP? Do we need a more consistent data formatting for these very small numbers? In terms of GWP, not worth the effort?

We have fixed the data formatting for the current version of PRIMAP-crf (2017-v1) and all data points are now formatted in scientific notation with data provided to 3 significant figures.

Kyoto bucket data (e.g. KYOTOGHGAR4) occur mixed in with the individual gas data? Although presentation of all data in one table with uniform formatting may prove necessary, the present organisation presents a substantial challenge to users (countries?) who might like to check specific PRIMAP outcomes e.g. where bucket total differs from sum of individual concentrations. Even if duplicative, should we have the bucket to individual components comparison as a separate table? Or the authors could build in a summary line before or after each bucket group, showing cumulative individual concentrations for comparison? We read about these discrepancies in the text e.g. in Section 3 but nowhere do we see a figure, which suggests to this reader that these discrepancies remain obscure except to the authors? Not worth the effort if the discrepancies remain minor but some countries will want to see?

Many of the discrepancies are not visible at the top categories for Kyoto-GHG because the absolute magnitudes of emissions involved are very small. Rather, they occur in the summing of lower level categories that we do not publish as part of PRIMAP-crf. The discrepancies are also, generally, still very small (<1%). To provide more transparency and explanation, we have added an additional table (CRF17-paper-data-discrepancytable) to the supplementary material that contains the relevant category and subcategory data for three of the country-specific issues described in section 3. The tables show the absolute and percent difference in emissions for the category and the summed sub-categories. During the data reading and preparation process, we found that it was necessary to go to the original data tables for each country to identify the cause of these discrepancies. The other issues described are either structural issues with the CRF tables, or are clear, specific errors (e.g. incorrect units).

We agree that the Kyoto-GHG data may be more widely used than the full dataset, and propose to make that data more readily available on our website (Paris-Reality-Check), but prefer to keep the main data repository as a single file. With many data processing tools, it should be relatively easy to filter the data for the categories and gases of interest.

In Figure 4, left panel LULUCF appears on rough estimate about cumulative sink of 20% or so. From global carbon budget also in ESSD, for decade 2007 to 2016, fossil fuel emission roughly 10 GtC with a net land impact (sink minus sources) of very roughly 2 GtC, so again roughly 20%. Thinking about validation, for which these authors address only the PRIMAP product vs the original CFRs. But they could perhaps validate their PRIMAP outcomes against carbon budget because the latter uses several sources in addition to the UNFCCC reports used here? Even a brief mention of validation in text would strengthen reader confidence in their PRIMAP outcomes?

As we understand it, you are here suggesting that we should cross-check the countryreported data against an independent estimate for further verification. Our goal with this study was not to verify that the country reported data is correct, but to make the dataset itself more accessible. In doing so, we want to ensure that we have not made mistakes in the processing and therefore make checks for internal consistency as this is an indicator that the data is correctly processed.

A possible use of this dataset is to further verify country reported data. Verification against the GCB is difficult as the data sources are slightly different, as are the categories of emissions. In the PRIMAP-hist dataset (Gütschow et al., 2016) we utilise several different emissions data sources (including CRF data) and in future work intend to include a comparison of these different sources.

We therefore consider the suggestion here out of the scope of this particular paper but will consider it for other analysis.

**Also in Figure 4, right hand panel about bunkers, we should interpret this more as who reports and who doesn't rather than as actual emission numbers?**

We don't think it's possible to interpret completeness of reporting from the data as shown. The wide spread in data results more from the fact that the share of emissions from each country is shown, thereby reflecting the geographic location and main industries of the countries, some of whom have a higher share of international shipping and aviation activities in their economies. Reporting of emissions from international bunkers is, however, somewhat complicated.

As with most activities, the IPCC Guidelines describe three options for estimating emissions from aviation and marine bunkers with increasing complexity (See Ch 3.6 and 3.X of IPCC 2006 Reporting Guidelines). The lower tiers for aviation are based on fuel sales data only, whereas the upper tier (3) is based on activity data (e.g. how many flights with which type of aircraft between which airport). Both tiers for shipping (water-born navigation) are based on fuel sales, with the upper tier incorporating country specific emissions factors and ship movement data where available.

The second challenge with reporting bunker emissions is in separating international and domestic activities, particularly when basing the reporting on fuel sales. The difference between domestic and international is based on the departure and arrival destinations (and not airline, ship ownership, or flag state).

The IPCC suggest a few methods for distinguishing between the two; taxation data where "fuel sold for domestic use is subject to taxation, but that for international use is not", or bottom-up data from airlines that may know the fuel consumption for domestic and international flights. The definition of 'international' and 'domestic' can lead to data discrepancies between country reported data and other sources.

Received and published: 12 May 2018

Thanks for the excellent paper and data. I have only minor comments, as the paper presents a well done analysis, including quite some technical details. The transparency of the results is very good.

Minor comments: 1/ I would also present the impact of the new reporting guidelines on the emissions relative to 1990 levels (of BY emissions), including and excluding LULUCF, in a table. The %-changes are presented on page 5, the second paragraph, but I would also present an overview in a Table.

Thank you for the suggestion, we think that it is a great idea of a use for this dataset and is a reason we produced it. However, this paper is about describing the data and how it was prepared and we prefer not to include analysis of the data. In section 3.2 of the manuscript we describe the change in average emissions over the full reporting timeperiod, rather than changes in emissions relative to the base year, that is change in trends when using alternate guidelines. The exercise you suggest is non-trivial (e.g. which year to plot relative to 1990, base year emissions for targets are not always the same as total reported emissions, etc.).

We will consider doing such analysis with this data in the future.

2/ Now the results of three countries are highlighted. I would also recommend to include a few additional other countries, as the current figure brings these countries on the spotlight.

The three displayed in this figure were specifically chosen as they are the only three countries that clearly have a shift in total LULUCF emissions associated with the change in guidelines. Reported historic LULUCF emissions change a lot for countries through time and for multiple reasons (e.g. updated activity data, changes in methodology, choice of which tier of reporting guidelines to use) and plots for other countries are therefore much more complicated. We therefore prefer not to add additional countries as they would rather raise issues with reporting in addition to the transition to the new reporting guidelines, which is the focus of this paper and dataset.

I think that this manuscript is well written and very useful for analyzing the large emission data and check possible errors in the reported data. I recommend this manuscript for publication.

Thank you for your positive feedback, we are glad that you find the dataset useful.

Overall, this is a great study, very detailed and sophisticated. Thanks for such a good work. This data can be used for many advanced research and publications. Recommended to be published in ESDD after few, but not major clarifications.

Thank you for your feedback and constructive comments below. We hope we have sufficiently addressed your points.

In the line 27th, it is stated that "Kazakhstan is included in the data set as a member of Annex I countries for the purpose of the Kyoto Protocol". What does this mean? Kazakhstan is considered as Annex-I because it ratified the Kyoto Protocol? If this is the reason, many other non-Annex I countries ratified Kyoto Protocol as well. Why only including Kazakhstan? Could you please clarify this little bit more?

The UNFCCC website notes on the download page for national inventory submissions and CRF data that:

"In accordance with the COP conclusion (FCCC/CP/2001/13/Add.4, section V.C.) and following ratification by Kazakhstan of the Kyoto Protocol on 19 June 2009 and its entry into force on 17 September 2009, Kazakhstan is considered an Annex I Party for the purposes of the Protocol but remains to be a non-Annex I Party for the purposes of the Convention."

The relevant COP conclusion states:

*"C. Amendment proposed by Kazakhstan to add its name to the list in Annex I to the Convention*2

1. At its 8 plenary meeting, on 9 November 2001, the Conference of the Parties, acting upon the recommendation of the Subsidiary Body for Implementation, took note that Kazakhstan, in accordance with Article 4, paragraph 2(g), had notified the Depositary on

23 March 2000 that it intended to be bound by Article 4, paragraphs 2(a) and (b) of the Convention. The Conference further noted that the Depositary had informed the other signatories and Parties of that notification, and that, upon ratification of the Kyoto Protocol by Kazakhstan and its entry into force, Kazakhstan becomes a Party included in Annex I for the purposes of this Protocol In accordance with Article 1, paragraph 7 of the Protocol.

2. The Conference of the Parties noted the interest expressed by Kazakhstan in engaging in negotiations with a view to defining a quantified emission limitation or reduction commitment for Kazakhstan under Annex B of the Protocol.

3. The Conference of the Parties recognized that Kazakhstan will continue to be a Party not included in Annex I for purposes of the Convention.

2 The heading of this item reflects the original request of Kazakhstan of 24 April 1999. The heading has been retained although this conclusion by the Conference of the Parties does not imply any amendment of the lists in the Annexes to the Convention."

We have added a reference to the COP decision to the manuscript, and the following additional sentence for further clarification:

"Kazakhstan voluntarily elected to be considered as an Annex I Party for the Kyoto Protocol, and thereby to have a quantified emissions reduction target and associated reporting obligations." (Page 2 | Lines 3-4)

Also, in the following sentence it is stated that Malta and Cyprus are reporting their emissions as members of European Union. Can you provide some reference for this, where do you get such information?

After joining the European Union, Malta and Cyprus agreed to be bound by the EU emissions reduction targets and enter into the EU Emissions Trading Scheme. They also were also officially added to the list of countries in Annex one of the Convention through amendments. The official decisions can be found on the UN website here: Cyprus - https://treaties.un.org/doc/Publication/CN/2012/CN.355.2012-Eng.pdf Malta - https://treaties.un.org/doc/Publication/CN/2010/CN.237.2010-Eng.pdf

We have updated the sentence in the original text to better reference and describe the situation. We also include a reference to additional countries that joined Annex I in the late 1990s. Malta and Cyprus are highlighted as unique because they joined Annex I and began reporting CRF data after the initial years of CRF data submission.

"Several Parties (Croatia, Cyprus, Czech Republic, Liechtenstein, Malta, Monaco, Slovakia, Slovenia) were not originally listed in Annex I but have subsequently been added through amendments to the Convention. Most recently, Malta (United Nations, 2010) and Cyprus (United Nations, 2012a) requested to be included in Annex I after joining the European Union (EU) and adopting the EU-wide emissions reduction targets under the Doha amendments (United Nations, 2012b)."

(Page 1| Line 26 to Page 2 | Line 1)

Lastly, at the data description file on page 2, it is stated there are 3 .csv files in the date sets. However, there are only 2 csv files in addition to two other text files. The following file is not there: "PRIMAP-crf-IPCC2006-category-codes.csv" Could you please include this file as well.

We apologise for this omission and now include the file in the data repository. The same file is also available as supplementary material to the paper.

Please also check the correction of the measurements, there are several variables/units, a mistake easily can be made. Since I am not able to go over each calculation in details, I can only suggest you making sure that there is no mistake in the calculations.

Thank you for noting the challenges with ensuring correct units and conversions. The PRIMAP emissions module itself has unit checks built-in to the framework and we further check the unit conversions, particularly to GWP baskets, by cross-checking against the country-reported calculations and also against the National Inventory Reports. This process is described in the Methods section of the paper (2.1).

After these corrections and clarifications, I recommend the study to be published in ESSD.

**PRIMAP-crf: UNFCCC CRF data in IPCC 2006 categories**

M. Louise Jeffery1, Johannes Gütschow1, Robert Gieseke1, and Ronja Gebel1

1 Potsdam Institute for Climate Impact Research (PIK), Member of the Leibniz Association, P.O. Box 60 12 03, D-14412 Potsdam, Germany

5 Correspondence to: M. Louise Jeffery (louise.jeffery@pik-potsdam.de)

Abstract. All Annex I Parties to the United Nations Framework Convention on Climate Change (UNFCCC) are required to report domestic emissions on an annual basis in a 'Common Reporting Format' (CRF). In 2015, the CRF data reporting was updated to follow the more recent 2006 guidelines from the IPCC and the structure of the reporting tables was modified accordingly. However, the hierarchical categorisation of data in the IPCC 2006 guidelines is not readily extracted from the

10 reporting tables. In this paper, we present the PRIMAP-crf data as a re-constructed hierarchical dataset according to the IPCC 2006 guidelines. Furthermore, the data is organised in a series of tables containing all available countries and years for each GHG individual gas and category reported. It is therefore readily usable for climate policy assessment, such as the quantification of emissions reduction targets.

In addition to single gases, the Kyoto basket of greenhouse gases (CO2, N2O, CH4, HFCs, PFCs, SF6, and NF3) is provided

- 15 according to multiple global warming potentials. The dataset was produced using the PRIMAP emissions module. Key processing steps include; extracting data from submitted CRF Excel spreadsheets, mapping CRF categories to IPCC 2006 categories, constructing missing categories from available data, and aggregating single gases to gas baskets. Finally, we describe key aspects of the data with relevance for climate policy; the contribution of NF3 to national totals, changes in data reported over subsequent years, and issues or difficulties encountered when processing currently available data. The processed
- 20 data is available under an Open Data CC BY 4.0 license, and available at http://doi.org/10.5880/pik.2018.001.

**1. Introduction**

Under the United Nations Framework Convention on Climate Change (UNFCCC), Annex I countries are required to report detailed GHG emissions inventories to the UNFCCC on an annual basis. All data are reported in the 'Common Reporting
Format' (CRF) Excel tables and accompanying National Inventory Reports (NIRs), which are made available on the UNFCCC website (UNFCCC, 2017c). Several Parties (Croatia, Cyprus, Czech Republic, Liechtenstein, Malta, Monaco, Slovakia, Slovenia) were not originally listed in Annex I but have subsequently been added through amendments to the Convention. Most recently, Malta (United Nations, 2010) and Cyprus (United Nations, 2012a) requested to be included in Annex I after joining the European Union (EU) and adopting the EU-wide emissions reduction targets under the Doha amendments (United

Nations, 2012b). In addition to those Parties now listed in Annex I of the Convention (UNFCCC, 1992), Kazakhstan is considered a member of Annex I for the purpose of the Kyoto Protocol (COP7 Decision FCCC/CP/2001/13/Add.4, section V.C.). Kazakhstan voluntarily elected to be considered as an Annex I Party for the Kyoto Protocol, and thereby has quantified emissions reduction targets and associated reporting obligations. Malta and Cyprus also report CRF data as members of the

5 European Union, but are not listed under Annex I of the Convention.

Prior to 2015, CRF data were reported following guidelines published by the IPCC in 1996 (IPCC, 1996), which detail the methodologies that should be used to calculate the inventories. Since 2015, reporting follows revised IPCC guidelines published in 2006 (IPCC, 2006). The methodologies to calculate emissions are primarily based on combining activity levels and emissions factors. For example, how many cows does a country have (activity), and what are the typical methane emissions

- 10 per cow (emissions factor)? For each source, multiple methods of calculation are possible with increasing complexity from tier 1 to tier 3. In general, tier 1 methods use a basic calculation and default emissions factors, tier 2 incorporates country specific emissions factors, and tier 3 may include advanced modelling approaches alongside country specific emissions factors. Tier 3 approaches are generally considered more accurate but may require more information and are more difficult to calculate. In addition to revisions in methodology and emissions factors, the IPCC 2006 guidelines (IPCC, 2006) and supplementary
- 15 methods arising from the Kyoto Protocol (IPCC, 2013) updated the range of activities and gases covered, and the hierarchical categorisation of the data. Major changes included (1) Combining categories 2 (Industrial Processes) and 3 (Solvent and Other Product Use) to the new category Industrial Processes and Product Use (IPPU) and (2) combining the Agriculture and Land-Use, Land-Use Change, and Forestry (LULUCF) categories into one Agriculture, Forestry, and Other Land-Use (AFOLU) category.
- 20 Although the more up to date methodological guidelines are now followed, the CRF reporting tables still closely resemble the structure of the previous version. For example, Agriculture and LULUCF emissions are still reported separately and top-level tables of fugitive emissions from fuels still reflect IPCC 1996 categories. The legacy effect is primarily a result of the negotiating process.

A consistent and complete hierarchical dataset according to the 2006 guidelines and categories allows (1) checks on data

- 25 consistency to be made, and (2) comparison with other datasets. For these reasons, we have extracted and processed all reported CRF data and re-organised it to the IPCC 2006 categorisation (Fig. 1). Furthermore, we make the data available in an easily used, machine readable, comma-separated values (CSV) file with aggregate gas baskets according to multiple global warming potentials (GWPs). Each table contains data for all countries and years available, allowing for time series analysis and comparison between countries. The data described and presented in this paper is that published in 2017 and available for
- 30 download by 15 December, 2017. CRF data is published annually in April and, where possible, that new data will be published in the same format and following the methods presented here.

[revised manuscript text omitted]

In the PRIMAP emissions module and in the published PRIMAP-crf data, we distinguish between the two sets of guidelines using separate sets of category codes. Data following the hierarchy of the IPCC2006 guidelines have category codes prefixed with 'IPC', and data following the 1996 categories have category codes with the prefixed 'CAT'.

**5. Data Availability**

15

20

The dataset is available from the GFZ Data Services under doi:10.5880/PIK.2018.001 (Jeffery et al., 2018). When using this dataset, or one of its updates, please cite this paper and the precise version of the dataset used. Please also consider citing the original UNFCCC source (UNFCCC, 2017c) when using this dataset. The data presented in this paper was released in 2017 and includes all data revisions until 15 December 2017.

New versions of the UNFCCC CRF data are released annually with an additional year of data. Some countries also submit revised versions of their data through the year. Where possible, the PRIMAP-crf data will be updated accordingly and a revised dataset released.

Data releases with an additional year of data are indicated in the naming of the data - the year of data publication is indicated

25 by the dataset name, e.g. PRIMAP-crf 2017-v1 data includes data first released by countries in 2017. Inclusion of subsequent data revisions from the same year are indicated by the version number, for example PRIMAP-crf 2017-v2 includes all CRF2017 data published by 1 June 2018.

**6. Conclusions**

[revised manuscript text omitted]

United Nations: C.N.237.2010.TREATIES-2 ADOPTION OF AMENDMENT TO ANNEX I TO THE CONVENTION IN ACCORDANCE WITH ARTICLE 16 (3) OF THE CONVENTION, [online] Available from: https://treaties.un.org/doc/Publication/CN/2010/CN.237.2010-Eng.pdf (Accessed 19 June 2018), 2010.

United Nations: C.N.355.2012.TREATIES-XXVII.7 ADOPTION OF AMENDMENTS TO ANNEX I TO THE
5 CONVENTION, [online] Available from: https://treaties.un.org/doc/Publication/CN/2012/CN.355.2012-Eng.pdf (Accessed 19 June 2018a), 2012.

United Nations: C.N.718.2012.TREATIES-XXVII.7.c DOHA AMENDMENT TO THE KYOTO PROTOCOL DOHA, 8 DECEMBER 2012 ADOPTION OF AMENDMENT TO THE PROTOCOL, [online] Available from: https://treaties.un.org/doc/Treaties/2012/12/20121217%2011-40%20AM/CN.718.2012.pdf (Accessed 19 June 2018b), 2012.

Figure 1: Top three levels of the IPCC 2006 categorisation. Filled circles indicate that further sub-categories exist. Orange circles indicate that the category is generated by aggregation of sub-categories in PRIMAP-crf. A fully expandable and explorable version
 of this graphic is available at <a href="https://www.pik-potsdam.de/paris-reality-check/primap-crf/">https://www.pik-potsdam.de/paris-reality-check/primap-crf/</a>.